# Non-Intrusive Maternal Style as a Mediator between Playfulness and Children’s Development for Low-Income Chilean Adolescent Mothers

**DOI:** 10.3390/children10040609

**Published:** 2023-03-23

**Authors:** Laura Léniz-Maturana, Rosa Vilaseca, David Leiva

**Affiliations:** 1Department of Cognition, Development and Educational Psychology, Universitat de Barcelona, 08007 Barcelona, Spain; 2Department of Social Psychology and Quantitative Psychology, Universitat de Barcelona, 08007 Barcelona, Spain

**Keywords:** adolescent mothers, playfulness, intrusiveness, children’s development

## Abstract

The aim of this study was to describe the relationship between low-income Chilean adolescent maternal playfulness and mothers’ non-intrusiveness in their children’s development and to analyze whether a mother’s non-intrusiveness mediates the relationship between maternal playfulness and children’s development. The Parental Playfulness Scale and the Subscale of Intrusiveness from the Early Head Start Research and Evaluation Project were used to assess maternal playfulness and mothers’ non-intrusiveness respectively. Ages and Stages Questionnaire 3rd Edition (ASQ-3) was applied to measure the children’s communication, gross and fine motor skills, problem-solving and personal–social development. The sample consisted of 79 mother–child dyads with children aged 10–24 months (M = 15.5, SD = 4.2) and their mothers aged 15–21 years old (M = 19.1, SD = 1.7). A bivariate analysis showed that maternal playfulness was significantly associated with communication, fine motor, problem-solving and personal–social development. Moreover, higher levels of communication, fine motor skills and problem-solving development were observed in the children of less intrusive mothers. Maternal playfulness had a significant effect on children’s development of language, problem-solving and personal–social skills when their mothers showed less intrusiveness during interaction. These findings contribute to the understanding of the interaction between adolescent mothers and their children. Active play and less intrusiveness can enhance child development.

## 1. Introduction

Adolescent mothers and their children have been studied by diverse authors due to the social problems that adolescent motherhood entails for the socioeconomic future of those mothers and for their children’s development outcomes [1,2,3,4,5,6,7]. Latin America has the second-highest number of adolescent mothers and motherhood in adolescence is more frequent in low-income countries [8]. Chile is one of the Latin American countries in which adolescent pregnancies have decreased [9]; however, these pregnancies are still mainly found among the poorest women in the country [10]. Biobío is one of the regions with the highest poverty rates in Chile, with a significant difference in the average poverty gap with respect to Chile’s other regions [11]. Meanwhile, the number of adolescent mothers whose rights have been infringed in this region is higher than that reported in Chile globally [12]. Therefore, it could be useful to determine the context of adolescent motherhood and child development in the Biobío region in Chile, since it has been shown that Chilean adolescent mothers need support in parenting [13] due to their lack of the sufficient maturity to face the responsibilities of motherhood [14,15,16].

The social and economic obstacles that adolescent mothers face explain why the maternal behavior of Chilean mothers is stronger in older women than in younger women [17]. Notably, it has been shown that low-income adolescent mothers have lower engagement and are more withdrawn with their children [18], which affects the parent–child relationship. From this perspective, origins in a socially disadvantaged sector characterized by poverty could explain why child development levels are lower than in children born to older mothers [1,19,20]. Furthermore, adolescent mothers tend to be unaware of the precise age at which certain developmental milestones should appear [3,21]. This affects the way they promote their children’s development and could lead to a lower level of cognitive, linguistic, and social development outcomes. Previous literature has shown that children born to adolescent mothers have greater socioemotional problems than adult mothers [5,6]. Likewise, in a study carried out by Firk [7], it was discussed that the cognitive development of children of adolescent mothers may be affected by the socioeconomic risk to which they are exposed in comparison with the children of adult mothers. However, the aforementioned authors affirm that positive maternal interactive behaviors can be important mediators and can have a significant effect on child development. In fact, it has been found that adolescent mothers under-stimulate verbal development, which explains why their children present lower levels of expressive and receptive language, attributed to high intrusiveness in dyadic interactions [22].

It is well known that maternal intrusiveness is characterized by excessive control of the child’s activities during an interaction, with mothers imposing their own will while children are playing and preventing their children from following their own pace [23]. On the one hand, high maternal intrusiveness has been associated with higher levels of children’s negative emotional behaviors [24] and lower levels of children’s cognitive development outcomes [25]. However, low intrusiveness can predict a higher level of children’s expressive language [26]. In contrast, playfulness, characterized by an interaction between mothers and children in which curiosity and creativity are promoted, has been related to better vocabulary skills and children’s emotional self-regulation [27]. In addition, according to Creaghe and Kidd [28], when children are playing make believe, a significant context is created for communicative development and sociocognitive processes. Scientific evidence has shown that adolescent mothers participate less in playing [18] and are more intrusive with their children than adult mothers [29]. For a better understanding of this issue, it is important to further investigate the concepts of maternal playfulness and intrusiveness and how they are related to child development.

### 1.1. Theoretical Framework

#### 1.1.1. Playfulness and Children’s Development

Playfulness is defined as parenting behavior whose willingness to play includes creativity and fun as components that stimulate cognitive development, generating enjoyment for the child [30]. Specifically, playfulness produces in children a state of mind that promotes reflective thinking, imagination, humor and curiosity [31]. Many studies have shown the impact of children’s playfulness on their development. In research carried out by Quinn et al. [32], it was shown that interactive play promoted imagination and the use of gestures between caregivers and children contributed to the development of the children’s communication skills, understanding and productive language [33]. Likewise, there is evidence that young children who perform pretend play are more likely to demonstrate higher skills in the development of executive functions [34]. Notably, playfulness contributes to children’s social and emotional development. It has been found that play decreases levels of emotional distress, is associated with self-regulation, and facilitates transitions to new experiences [35]. One possible cause of the association between playfulness and social development is that children interpret and represent roles that lead them to regulate their own behavior to adapt to different contexts and contribute to expanding their vocabulary and narrative skills [36]. The link between playfulness and language development can be seen in the relationship between babbling and complex symbolic action [37], that is, the development of thought through images, symbols and the representation of ideas with gestures and vocalizations. For this reason, the game benefits children’s learning in terms of acquiring greater vocabulary, improving expression and understanding language that allows them to acquire new words [28].

Although previous research has been based on the interconnection between playfulness and cognitive, linguistic and social development, there is also evidence of a direct relationship between play and children’s motor development. According to Trevlas [38], playfulness and motor skills could be associated because movement is a natural action in early childhood, one that considers the learning and development expressed in playful activities. However, most of the studies that have analyzed associations with playfulness that include movement—some of them observed in mother–child interactions—have been carried out with samples of children with disabilities [39,40,41].

Considering the close correlation between playfulness and child development and its beneficial implications [42], existing knowledge should be extended by focusing on adolescent mothers, since their situation might be challenging for the reasons outlined below. Various authors have stated that low socioeconomic status in adolescent mothers interferes with the quality of the dyadic interaction between mothers and their children, which affects child development [7,15,16,23]. In a context of high indices of social and economic vulnerability, such as those experienced by the majority of Chilean adolescent mothers, there is less probability of providing cognitive stimulation according to the children’s characteristics and needs. This may result in negative parenting [43].

From this perspective, maternal playfulness has been associated with a higher level of warmth and sensitivity on the part of the mother [44]. A high level of intrusiveness has been linked to a low level of maternal playfulness [31]. It is well known that intrusion is negatively associated with a mother’s warmth, sensitivity and the use of positive regard towards their children [45,46]. These qualities are positively correlated with children’s reading and writing skills in later years [47] and with children’s social behavior [48]. However, despite the association between high maternal sensitivity and low maternal intrusiveness, mothers may display aspects of both in varying amounts depending on the context, so it is crucial to analyze low maternal intrusiveness independently of maternal sensitivity in a playful interaction, as in the Broomel study, to analyze its association with children’s development outcomes [49].

#### 1.1.2. Maternal Intrusiveness

As Ainsworth et al. [50] originally explained, intrusion is behavior in which mothers do not respond to their children’s cues and interests and do not promote infant autonomy and maternal sensitivity. This leads to interruptions and actions that focus on their own goals and not those of their children. In addition, intrusiveness is characterized by behavior in which mothers are overly controlling and overly involved during the interaction with their child [51]. In contrast, a mother’s non-intrusiveness refers to her ability to be available to her child by avoiding excessive stimulation or interfering with her child’s autonomy [31]. It has been found that Latino parents tend to be more authoritarian, controlling and overemphasize obedience and respect as important parenting factors. These variables are considered priorities for socialization [52]. Hence, a mother’s non-intrusion has been associated with better social levels of the child [53]. Inversely, negative, intrusive parenting is associated with lower performance in children’s cognitive development [26] and communication. A disruption in language learning can inhibit children with ambiguous or infrequent communication, which leads to a loss of opportunities for a child’s response [54]. It can be appreciated that intrusiveness may affect various areas of children’s development, especially social skills [55,56,57].

However, unlike playfulness, the evidence of an association between motor development results and maternal intrusiveness is scarce. Fine motor development has been associated with positive parenting behavior [58]. Therefore, it would be interesting to analyze whether low-income adolescent mothers’ non-intrusiveness is an important mediator of playful interaction with their children and their developmental outcomes, since maternal intrusion has generally been associated with negative child behavior [31,59]. As mentioned above, a mother’s non-intrusiveness avoids interfering with child autonomy [31], while parental behaviors that promote a child’s autonomy are related to adaptive child development, children’s executive function, sustained attention and emotional development [60,61,62]. In young mothers, it has been found that, depending on psychosocial factors, they may be more likely to over-stimulate and interrupt when a child is playing. They are prone to focus on their own interests and to prohibit their child’s activities [63]. Despite this, to the best of our knowledge, there have been no studies examining the impact that maternal playfulness may have on the development of Chilean children born of adolescent mothers, when they are non-intrusive in a dyadic interaction with their children. This gap in knowledge is relevant, particularly because previous research has addressed the impact of parenting intrusiveness on low-income children’s development in play interactions [24,26,64]. In turn, due to the high risk of vulnerability among adolescent mothers, it is useful to study aspects that contribute to supporting their parental behavior in order to optimize the development of their children [65]. Adolescent mother–child dyads form part of a group that requires special attention in Chile, since it has been shown that exposure to an environment of high social and economic vulnerability leads to a lack of cognitive stimulation, which affects children’s learning development [44,66].

### 1.2. Current Study

This study aimed to describe maternal playfulness and Chilean adolescent mothers’ non-intrusiveness during free play situations at home with typically developing children. The objective was to analyze the relationship between these maternal behaviors and their children’s development. The specific aims were to analyze the relationship between low-income Chilean adolescent mothers’ playfulness and non-intrusiveness and their children’s communication, gross and fine motor skills, problem-solving and personal–social development; and to explore whether mothers’ non-intrusiveness mediates the relationship between maternal playfulness and their children’s development. Therefore, the current study investigated the following research questions and hypotheses.

Are maternal playfulness and mothers’ non-intrusiveness related to their children’s development in Chilean dyads? We hypothesize that higher scores for maternal playfulness would be positively associated with a lower level of non-intrusiveness in adolescent mothers. In turn, this would be significantly related to children’s greater communication, problem-solving and personal–social development.Does Chilean adolescent mothers’ non-intrusiveness mediate maternal playfulness and is it related to their children’s development? We expect a mediated relationship of maternal playfulness and development via non-intrusiveness. In this regard, we hypothesize a negative relationship between playfulness and intrusiveness, which in turn will be negatively associated with children’s developmental outcomes.

## 2. Materials and Methods

### 2.1. Study Design

The current study was carried out using a descriptive observational approach (i.e., a correlational approach) based on a cross-sectional design. The participants were 42 boys (53%) and 37 girls (47%) aged two years or younger (10–24 months) (M = 15.5, SD = 4.2) with mothers aged from 15 to 21 years old (M = 19.1, SD = 1.7). The eligibility criteria were as follows: mothers who had become pregnant at 19 years old or younger, and children with typical development aged from 10 to 24 months.

### 2.2. Recruitment to the Study

Children and their mothers were recruited from seven family health centers, four day care learning centers, one residential home that receives homeless adolescent mothers, and one hospital, from provinces in Biobío Region in Chile. The coordinators of these institutions were contacted by e-mail and telephone and informed of the nature of the study. Subsequently, after being asked to screen the dyads’ eligibility, they provided personal information that included telephone and e-mail to contact the participants, and the dates of birth of children and their mothers, in accordance with the inclusion criteria. Mothers who agreed to participate were visited at home and were informed of the nature of the research. Those aged 18 or older signed an informed consent form and those below this age an informed assent form; in the latter case, their legal guardians provided an informed consent.

### 2.3. Procedure

One researcher visited all recruited dyads at home. Firstly, the ASQ-3 was applied to assess whether children performed the behavior indicated in the items of the areas of the instrument. The following instruction was given: “We will read each question and verify whether your child performs the activity, performs it sometimes, or doesn’t perform it yet”. ASQ-3 was completed in around 10–15 min. Subsequently, mothers were invited to play with their infants for 10 min and were informed that the session was recorded. The following instruction was given in Spanish: “Interact and play with your children as you typically do”. Then, videotaped observations of mother–child interactions were obtained during 10 min of free play using the “Three bag task” [67] that included books, toys for pretend play, and age-appropriate manipulative toys such as teddy bears, dolls, building blocks, cars and puzzles. The Parental Playfulness Scale [31] and the Subscale of Intrusiveness from the Early Head Start Research and Evaluation Project [68] were then used to score mother–child interactions in these videotaped sessions, which were later coded in the researchers’ computers. Figure 1 shows the number of participants who were initially selected for the analysis of the results, the reasons for exclusions, and the final number of participants.

### 2.4. Ethical Statement

The study was approved by the Health Service of Concepción and the Health Service of Biobío, in accordance with the Declaration of Helsinki (1964 and subsequent updates) and the Guidelines for Good Clinical Practice (GCP).

### 2.5. Instruments

#### 2.5.1. Parental Playfulness Scale

Maternal playfulness was coded using the Parental Playfulness Scale [32] through the observation of free play activities in 10-min videos. This tool evaluates the degree to which parents display creativity, curiosity, imagination and simulation when interacting with their children in a play context. The instrument is coded on a 7-point Likert-type scale rated from 1 (no playfulness) to 7 (high levels of creative play). Specifically, a score of 1 is given when it is not observed beyond brief verbal directives, or if mothers do not play, show no signs of creativity, or do not interact. A score of 2 means that parents spent most of the time labeling objects and there are few instances of concrete play. A score of 3 indicates that parents give commands about how to use the toys in a conventional manner. A mid-scale score of 4 is coded if half of the playful interaction is concrete and the other half is imaginary. A score of 5 indicates that imaginary play takes up most of the time of interaction. A score of 6 is coded when the mother adds creativity to the interactions, with imaginary play being an important quality and 7 means that parents spend most of the time in pretend play using the toys in an unconventional way. Two trained investigators jointly coded the videotapes. Inter-rater reliability for this research was calculated from 25% of observations, yielding an inter-rater agreement estimate of 0.85. In this sample, observed maternal playfulness ranged from 1 to 6 (Mdn = 3.0, IQR = 3.0).

#### 2.5.2. The Subscale of Intrusiveness from Early Head Start Research and Evaluation Project

The Subscale of Intrusiveness from the Early Head Start Research and Evaluation Project: Child–Parent Interaction Rating Scales for the Three-Bag Assessment [68] was used to assess the mothers’ non-intrusiveness. Intrusiveness is defined as the control exerted by mothers focused on themselves instead of recognizing the needs and interests of their child. The item is scored on a 7-point Likert scale from 1 (very low intrusiveness) to 7 (very high intrusiveness). A score of 1 means that there are no signs of intrusive behavior in any maternal behavior. A score of 2 indicates low intrusiveness in which only a few instances of intrusive maternal behavior are observed. A score of 3 shows moderately low intrusiveness and mothers may initiate some interactions with their children that are not welcome, as shown by the children protesting or responding defensively to their mothers. Mothers may continue the activity after their children respond defensively. A score of 4 refers to moderate intrusiveness, that is, mothers may intrude abruptly on the child or show mild intrusiveness during the play interaction. A score of 5 means moderately high intrusiveness in which children have little opportunity during the play interaction to do anything on their own. A score of 6 corresponds to high intrusiveness in which mothers strongly deny the children the opportunity to perform activities by themselves. In this case, the children have few opportunities to experience autonomy. A score of 7 is very high intrusiveness and it means that the majority of the play interaction time is marked by the mothers completely controlling and allowing the children almost no self-direction in their activities. In the current study, 25% of mother–child interactions were coded by two trained observers. Inter-rater reliability agreement was 0.80. In this sample, the mothers’ intrusiveness score was from 1 to 5 (Mdn = 2.0, IQR = 1.0).

#### 2.5.3. Ages and Stages Questionnaire

The Spanish version of the Ages and Stages Questionnaire—3rd Edition (ASQ-3) [69] is a caregiver-completed questionnaire including 30 questions from 5 areas: communication (comprehensive and expressive language), gross motor skills (proximal movements and big muscle groups), fine motor skills (coordination and movements that include small and distal muscles), problem-solving (cognitive skills that include play and learning), and personal–social skills (a child’s self-help skills and interactions with others) based on milestones that should be achieved from birth to 60 months. Each area has 6 items, and for each item one of three possible responses is coded: 10 (yes), 5 (sometimes) or 0 (not yet). The response must be chosen depending on whether the child can perform a certain task. The score is obtained from the sum of their responses, and higher scores indicate better children’s development. The questionnaires were divided by the child’s biological age, given that there are series of questionnaires appropriate for different age ranges. Specifically, each child must be evaluated according to their biological age, which is the criterion for choosing the appropriate questionnaire. Thus, in this study, children’s developmental scores were transformed into Z scores for a common metric and differences in cut-off scores were avoided according to age ranges established in the instrument. For this sample, it was decided to have the questionnaire completed by mothers, but it was read and all items were checked by a member of the research team that visited them at home, as recommended by Small [70], to prevent any comprehension difficulties resulting from the mothers’ low literacy level. ASQ-3 has been validated in Chile [71] and has been recommended by UNICEF. In multinational studies, it has shown reliability = 94%, sensitivity = 88% and specificity = 82.5% [72]. In this sample, reliability was demonstrated by a Cronbach’s α of 0.73.

### 2.6. Data Analysis

Statistical analyses were calculated to describe the sample and the empirical distribution of the relevant variables in the study.

To address the first research question linked to the relationship between maternal playfulness, mothers’ non-intrusiveness and children’s development (communication, gross motor, fine motor, problem-solving and personal–social), non-parametric tests were carried out based on Kendall’s correlations. Additionally, 95% confidence intervals for all association indices were estimated using BCa Bootstrap with 5000 resamples.

Regarding the second research question, mediation models based on regression as described in Hayes [73] were estimated to assess the mediator role of mothers’ non-intrusiveness in the relationship between maternal playfulness and children’s development. More specifically, five mediation models were conducted to examine the mediation process for each dependent variable corresponding to the areas of ASQ-3 (communication, gross motor, fine motor, problem-solving and personal–social development). All the hypotheses about the pathways (i.e., direct and indirect effects) involved in the mediation models were tested using 95% BCa bootstrap CIs with 5000 replications. Statistical analyses were carried out using CI PROCESS [73] IBM SPSS version 27 for Windows, and R version 4.2.2.

## 3. Results

Seventy-nine dyads participated in this study. Specifically, fifty-six children resided with their mothers and grandparents (71%), seventeen children resided with their mother, father and grandparents (21%), and only 8%, corresponding to six children, resided with their fathers and mothers. Most mothers received childcare support provided by their children’s grandmothers (78%, n = 62) and only 28 children (35%) attended a preschool center. More than half of the mothers had completed high school (62%, n = 49), 29% had completed primary school (n = 23), and less than half of the mothers had not completed primary (9%, n = 7). From this group, 14% corresponds to mothers who dropped out of school due to maternity. Most of the mothers were not working (81%). On average, families reported a monthly income of 509.2 (USD), ranging from 108 to 678 (USD) (SD = 171.3). Thus, considering the criteria of the Chilean Association of Market Researchers 2019 [74] this sample was composed of mid- to low-income (678.49 USD or less) and low-income (391.16 USD or less) dyads. The results of this study are presented below.

Research Question 1: Are maternal playfulness and Chilean adolescent mothers’ non-intrusiveness related to their children’s development?

Table 1 shows the mean, SD, minimum and maximum ASQ-3 raw scores for each area of children’s development. In general, relatively low scores can be seen, except for personal–social development, considering the interval ranges of the instrument expected of children with typical development.

As reported in Table 2, the scores of the Parental Playfulness Scale and the Subscale of Intrusiveness were negatively related. The strength of the relationship was moderate (Τ = −0.4, *p* < 0.001; 95% CI = (−0.548; −0.232)). Moreover, maternal playfulness was significantly correlated to children’s communication (Τ = 0.3, *p* < 0.001; 95% CI = (0.082; 0.409)), fine motor (Τ = 0.2, *p* = 0.045; 95% CI = (0.014; −0.313)), problem-solving (Τ = 0.2, *p* = 0.041; 95% CI = (0.029; 0.328)), and personal–social development (Τ = 0.2, *p* = 0.005; 95% CI = (0.074; 0.374)). In turn, communication (Τ = −0.3, *p* = 0.004; 95% CI = (−0.408; −0.068)), fine motor (Τ = −0.2, *p* = 0.009; 95% CI = (−0.384; −0.067)) and problem-solving development (Τ = −0.3, *p* = 0.002; 95% CI = (−0.399; −0.101)) were negatively associated with mothers’ intrusiveness.

Research Question 2: Does Chilean adolescent mothers’ non-intrusiveness mediate maternal playfulness and its impact on their children’s development?

Table 3 includes all estimated arcs of the models for maternal playfulness on the five children’s development areas (communication, gross motor, fine motor, problem-solving, and personal–social development) mediated by mothers’ intrusiveness. Figure 1, Figure 2, Figure 3, Figure 4 and Figure 5 specify the direct effects of maternal playfulness on the mediating variable: mothers’ intrusiveness (a1). Moreover, they indicate the effects of the mediating variable (mothers’ intrusiveness) on the child development area of ASQ-3 (b1). The c’ path represents the direct effect of maternal playfulness on children’s development, while controlling for the mediated effects. Notably, in the mediation model, the total effect of maternal playfulness on children’s development (i.e., the c term) was estimated as the sum between estimates c’ and the indirect effect (a1b1).

Model 1 showed that the direct effect of maternal playfulness (c′ = 0.14, *p* = 0.08; 95% CI = (−0.02–0.30)) and the indirect effects on children’s communication mediated via mothers’ intrusiveness (a_1_b_1_ = 0.08; 95% BCa-CI = (−0.0007–0.15)) were non-significant. However, the total effect yielded a significant result (c = 0.21; 95% CI = (0.07–0.35)). This model accounted for 14.0% of the variance of the ASQ-3 communication development scores (see Figure 2).

Model 2 indicated that the direct effect between maternal playfulness and children’s gross motor development was not significant (c′ = 0.11, p = 0.19; 95% CI = (−0.06–0.28)) for the indirect effects via mothers’ intrusiveness (a_1_b_1_ = −0.06; 95% BCa-CI = (−0.30–0.17)) or for the total effect (c = 0.13; 95% CI = (−0.01–0.28)). Moreover, this model had low predictive capacity, accounting for 4.0% of the variance. Therefore, it cannot be considered useful for predicting children’s gross motor development (See Figure 3).

Model 3 shows that the direct effect between maternal playfulness and children’s fine motor development was different to zero but not significant (c′ = 0.07, *p* = 0.37; 95% CI = (−0.09–0.24)) Nevertheless, for indirect effects via mothers’ intrusiveness (a1b1 = 0.07; 95% BCa-CI = (−0.004–0.16)) and total effect (c = 0.15; 95% CI = (0.001–0.29)) the results were significant. Predictive capacity accounted for 9.0% of the variance of the ASQ-3 score’s fine motor area (See Figure 4).

The direct effect between maternal playfulness and children’s problem-solving indicated in Model 4 was not significant (c′ = 0.07, *p* = 0.36; 95% CI = (−0.09–0.23)). However, the indirect effects via mother’s intrusiveness (a_1_b_1_ = 0.09; 95% BCa-CI = (0.02–0.19)) and the total effect (c = 0.16; 95% CI = (0.02–0.31)) were significant. This model had a predictive capacity that accounted for 13.0% of the variance of the ASQ-3 problem-solving development scores (See Figure 5).

Both the direct (c′ = 0.24, *p* = 0.004; 95% CI = (0.08–0.40)) and the total effect (c = 0.22; 95% CI = (0.08–0.35)) between maternal playfulness and children’s personal–social development specified in Model 5 were significant. However, the indirect effect via mother’s intrusiveness (a_1_b_1_ = −0.23; 95% BCa CI = (−0.10–0.045)) was not significant. This model presented a predictive capacity that accounted for 11.0% of the variance of the ASQ-3 personal–social development scores (See Figure 6).

## 4. Discussion

The purpose of this study was to analyze the relationship between low-income Chilean adolescents’ maternal playfulness and mothers’ non-intrusiveness, and their children’s development. The objective was also to examine whether children’s development could be predicted by maternal playfulness when their mother’s intrusiveness was lower. Our findings indicate that there is a close association between maternal playfulness and mothers’ intrusiveness. This result is similar to a study carried out by Menashe and Atzaba-Poria [30] in which maternal playfulness was assessed with the Parental Playfulness Scale [31] in low-income adult mothers and was related to lower levels of intrusiveness. Additionally, the authors found that mothers, on average, showed moderate levels of playfulness. A similar result was found in a study carried out by Cabrera et al. [27]. As expected, the mean score for playfulness in the sample of the current study was lower than in the other previously indicated studies. This result can be explained as it is well known that the quality of adolescent mother–child interactions may be affected by several demographic challenges associated with the poverty that they face [74]. This leads to adolescent mothers having greater difficulty in bonding with their child [75] and being less involved in playing with their children [18]. In turn, the difficulties they face may lead to emotional problems in adolescent mothers that could imply more intrusive mother–child interactions [76]. Interestingly, the level of intrusiveness of adolescent mothers in this sample was relatively low, in contrast to previous studies that have found higher levels of intrusiveness in adolescent mothers [29]. Usually, demographic risks could influence higher levels of mothers’ intrusiveness [77]. Although this sample was composed of low-income dyads, we did not analyze risk variables such as maltreatment, abuse or neglect that typically form part of this vulnerable group [78] and which are associated with higher levels of maternal intrusiveness [79]. Thus, these factors should be examined in future studies of adolescent mothers in order to assess the possible impact of the quality of support in childcare, since in this sample most of the adolescent mothers reported receiving support from their family. However, this variable was not assessed.

We also tested whether maternal playfulness and mothers’ non-intrusiveness were related to their children’s development. Our results show that mothers who were more playful and less intrusive had children with higher levels of communication, fine motor and problem-solving development than mothers who were less playful and more intrusive. These results are in line with Yogman et al. [80], who indicated that a nurturing relationship between parents and children can be better established through play, to optimize children’s cognitive, language and social skills. Particularly, previous evidence has demonstrated that parental play interaction has a significant impact on children’s cognitive development [81] and pretend play has a positive effect on the executive functions of children, unlike other types of play [82]. In this context, imaginative play may be considered a positive instance of developing executive functions [83], as play encourages children’s imagination and memory, and is essential for the development of thinking [84]. Cognitive skills such as executive function and attentional control are also linked to lower levels of mothers’ intrusiveness [85], which is concordant with our results. This aspect is reasonable, because maternal intrusiveness tends to be negatively related to free play and predicts lower levels of children’s attention [86].

It should be noted that the fact that pretend play is related to cognitive skills, including symbolic understanding [87], could explain why we found that maternal playfulness was also associated with children’s communication development. The relationship between play and language development is in the ability of children to use and understand symbols [88], which shows that symbolic play is significantly linked to expressive and receptive language. In fact, in the study carried out by Cabrera et al. [27], the authors indicate that the advantage of play is that it encourages children to learn to understand and say new words. As in our results, they found that parental playfulness was related to language, specifically in mothers when fathers showed higher levels of parental playfulness. Moreover, our results are also similar to other studies, which found that when mothers and fathers had higher levels of intrusiveness during play interactions with their children, the children showed lower levels of expressive language [51,89]. We did not analyze father–child interactions and their relationship with children’s development, given that in the Chilean context some adolescent fathers do not reside with their child [90], as in our sample. However, our findings add to the literature by providing evidence that adolescent mothers may enhance their children’s development when playful and less intrusive interactions with their children are promoted.

We found that maternal playfulness was related to fine motor skills but not gross motor skills. Even though gross locomotion movements are used in children’s play, they are usually referred to as physical play (i.e., walking, crawling and rolling) [91], a variable that we did not analyze in the interaction between mothers and children in our study. However, both maternal playfulness and mothers’ non-intrusiveness were related to children’s fine motor development. In parent–child interactions, as measured by three-bag tasks, it has been found that fine motor skills can be improved when there is high-quality early parenting [92]. Under this perspective, this study provides evidence of the importance of playfulness and non-intrusiveness on children’s motor development to the increasing literature about parenting in adolescent mothers.

Regarding the relationship between maternal playfulness and personal–social development, which refers to a child’s self-help skills and interactions with others, our results were concordant with a study carried out by Choi and Kim [93] that found that playfulness influences children’s social skills. It is well known that when children are playing, they assume different roles and thus learn about social interactions [94]. In this sense, our results indicate that maternal playfulness in adolescent mothers is generally an important factor in children’s personal–social development. This could occur because when parents accept that children are active protagonists in their play, then they have autonomy over their activities. This is one of the most empowering experiences a child may have [95] and one which enhances their social and self-help skills. Interestingly, mother’s intrusiveness was not significantly linked to better personal–social development in this research. In contrast, some studies have concluded that intrusive parenting is associated with lower levels of social abilities [96,97]. Nevertheless, previous literature that has studied adolescent motherhood has found that maternal mental health is a relevant factor in the social and emotional development of their children [98,99], which could be a key indicator to further analyze the relationship of personal–social development with maternal factors in Chilean adolescent mothers. Despite this fact, mothers’ non-intrusiveness was an important mediator in the relationship between maternal playfulness and children’s development, including personal–social development in this study.

The findings of our study indicate that maternal playfulness in adolescent mothers has a significant effect on most measured levels of child development, notably communication, fine motor, problem-solving and personal–social development. However, despite the fact that, in general, the association between maternal playfulness and children’s development was seen to be relatively low, the mothers’ non-intrusiveness played a buffering role. This finding aligns with previous research that has shown the negative association between parental intrusiveness and child development [22,30,60,61,62]. Thus, our results highlight the importance of examining intrusiveness in contexts of mothers at high risk of social and economic vulnerability. The youngest Chilean mothers and their children are a particularly interesting field of study, because it has been demonstrated that parental skills in mothers are relatively lower than in older mothers with higher socioeconomic status [17].

### Limitations and Future Directions for Research

The present study extends the current literature on adolescent mothers’ interactions with their typically developing children. Nevertheless, the study has some limitations that should be considered when the results are interpreted. First, the sample was not probabilistic. The procedure used to recruit the mothers in this sample may have been conditioned by the willingness to participate. [100]. Conceivably, the mothers who agreed to participate in this study were probably the most informed and the most involved in childrearing.

The small sample size may have compromised the statistical power of the tests used in the present study. However, power analyses showed that a sample size of 84 participants would be necessary to detect moderate correlations (r = 0.3, significance = 0.05, and power = 0.80). In addition, post-hoc power analyses for a sample size of 79 participants showed that the statistical power for detecting moderate bivariate associations was 0.77. Given the potential issues related to statistical power, bias-corrected bootstrap confidence intervals were used to reliably estimate indirect effects, as recommended by other authors [101].

A main limitation, given its cross-sectional and correlational nature, is that causality cannot be inferred from the main findings of the current study. Nonetheless, association patterns between ASQ-3 scores and maternal styles are necessary for their understanding. In contrast to previous research that has analyzed parental playfulness assessed with the Parental Playfulness Scale [18,27,28,29,30], the current study did not consider the quality of parental interaction between fathers and their children. The study could have been more robust if we had complemented and compared fathers’ playfulness and the intrusiveness of adolescent mothers. However, most of the adolescent mothers were not cohabiting with the father of their children. As mentioned previously in the discussion section, a high number of Chilean adolescent fathers do not live with their child [89]. Additionally, the development of children born of adolescent mothers may be enhanced by the presence of grandparents when the father is absent [102], since Chilean adolescent mothers need support in childcare [13,15,16,103]. It has been shown that a large number of Chilean grandmothers share the responsibilities in the provision of care for children [104]. Though the participation of the grandparents could have revealed important information, some adolescent mothers receive poor or no social and psychosocial support from their families [105]. Actually, our sample was composed of some dyads that did not cohabit with fathers or grandparents. Despite this fact, including fathers and grandparents could be fruitful for future research of this nature.

Finally, the instruments used to measure dyadic interactions between mothers and children have not been previously applied to a Chilean sample. However, intrusiveness and playfulness have been studied in samples of dyads in low-income Latin American mothers and children [27,72], and in samples of Spanish-speaking mothers and children [106]. All these studies used the same instruments as the ones used here, suggesting that they are suitable for use with this Chilean sample, given the similarities in culture and ethnic group.

Given all the issues mentioned above, we stress the need for further research using probabilistic methods and larger samples as well as longitudinal designs to capture the causal nature of the mechanisms relating parental styles and children’s development and to replicate the effects found in the present study.

## 5. Conclusions

This study confirmed significant findings on the effect of maternal playfulness focused on creativity, imagination, humor and curiosity on the development of children of Chilean adolescent mothers. Although the effects found were low to medium sized, it is important to promote low-income adolescent mothers’ playfulness in their interactions. This promotes autonomy in their children and avoids maternal intrusiveness, since our results indicate that low-income mothers’ intrusiveness could act as a protective factor when mothers play with their children. However, it is important to emphasize the importance of improving maternal playfulness and considering that the sample studied is part of a group with greater social and economic vulnerability. This should receive significant attention in Chilean national intervention programs. Therefore, our findings highlight a growing understanding of maternal playfulness and mothers’ intrusion that could help in the effectiveness of intervention programs that contribute to reduce developmental problems in children born of Chilean adolescent mothers. In conclusion, these findings contribute to offering information about important characteristics of interaction between adolescent mothers and their children. Demonstrating active play and less intrusiveness offer opportunities for adolescent mothers to enhance both the interaction between them and their children and their children’s development.

## Figures and Tables

**Figure 1 children-10-00609-f001:**
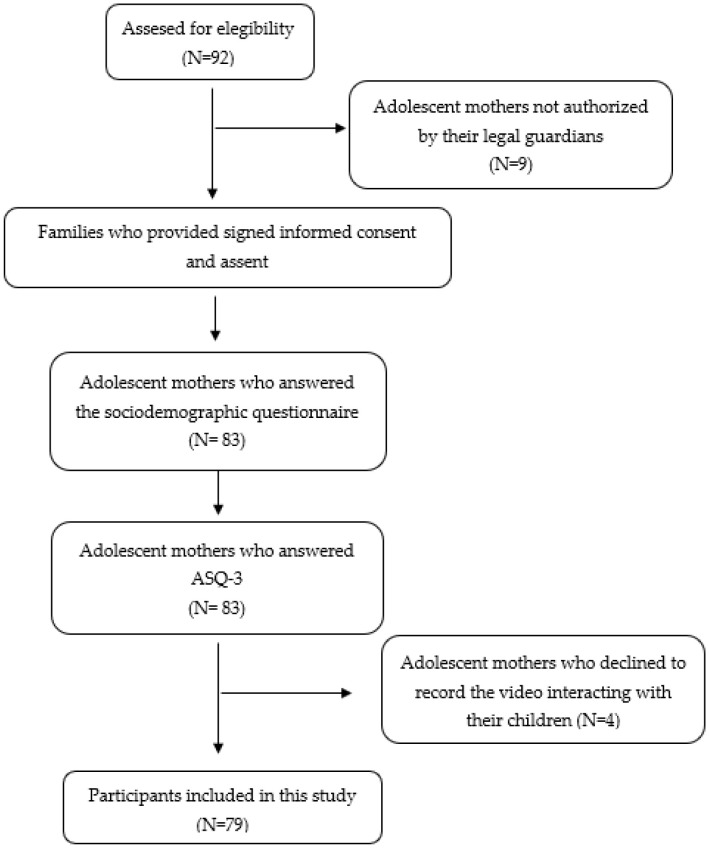
Participant flowchart.

**Figure 2 children-10-00609-f002:**
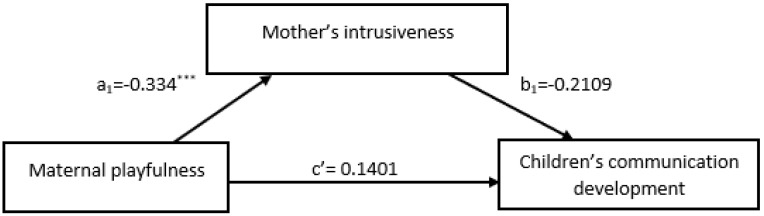
Effect of maternal playfulness on children’s communication development mediated by mothers’ intrusiveness. Asterisks (***) indicate the significance effect of maternal playfulness on mother’s intrusiveness (*p* < 0.001) (a1).

**Figure 3 children-10-00609-f003:**
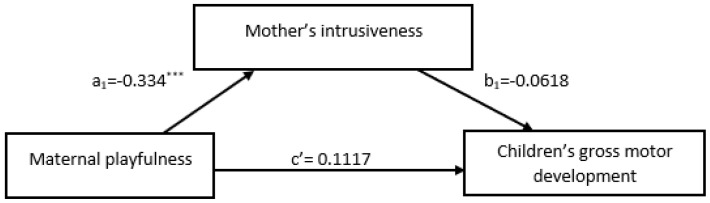
Effect of maternal playfulness on children’s gross motor development mediated by mothers’ intrusiveness. Asterisks (***) indicate the significance effect of maternal playfulness on mother’s intrusiveness (*p* < 0.001) (a1).

**Figure 4 children-10-00609-f004:**
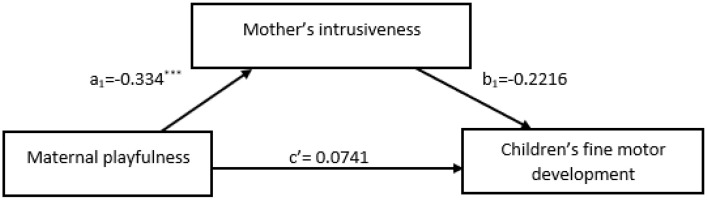
Effect of maternal playfulness on children’s fine motor development mediated by mother’s intrusiveness. Asterisks (***) indicate the significance effect of maternal playfulness on mother’s intrusiveness (*p* < 0.001) (a1).

**Figure 5 children-10-00609-f005:**
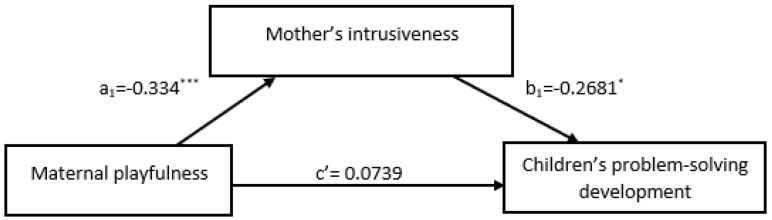
Effect of maternal playfulness on children’s solving-problem development mediated by mother’s intrusiveness. Asterisks (***) indicate the significance effect of maternal playfulness on mother’s intrusiveness (*p* < 0.001) (a1), and asterisk (*) indicates the significance effect of mother’s intrusiveness on children’s problem solving (*p* < 0.05) (b1).

**Figure 6 children-10-00609-f006:**
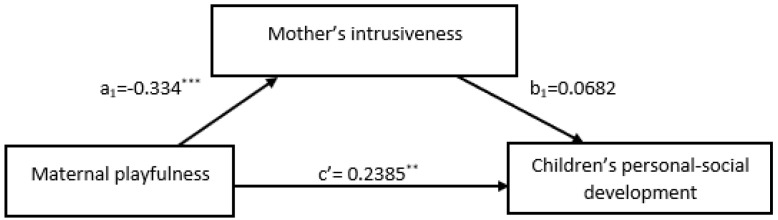
Effect of maternal playfulness on children’s personal–social development mediated by mother’s intrusiveness. Asterisks (***) indicate the significance effect of maternal playfulness on mother’s intrusiveness (*p* < 0.001) (a1). And asterisks (**) indicate the direct significance effect of maternal playfulness on children’s personal-social development (*p* < 0.01) (c′).

**Table 1 children-10-00609-t001:** Descriptive data on children’s development ASQ-3 scores.

	Min.	Max.	M	SD
Communication children’s development (0–60)	0	60	29.49	11.59
Gross motor children’s development (0–60)	5	60	39.24	16.27
Fine motor children’s development (0–60)	10	60	37.41	12.24
Problem-solving children’s development (0–60)	0	60	37.41	12.22
Personal–social children’s development (0–60)	20	60	41.46	10.23

Note: Raw values of the children’s development.

**Table 2 children-10-00609-t002:** Kendall’s correlations (Τ) between maternal playfulness, intrusiveness scores, and children’s ASQ-3 scores. Bootstrap BCa 95% CIs are shown within brackets.

	1	2	3	4	5	6	7
1 Parental Playfulness Scale		−0.400 **	0.253 **	0.146	0.170 *	0.173 *	0.237 **
2 Subscale of Intrusiveness			−0.248 **	−0.108	−0.227 **	−0.266 **	−0.088
3 Communication ASQ-3 area				0.244 **	0.257 **	0.231 **	0.247 **
4 Gross motor ASQ-3 area					0.280 **	0.273 **	0.214 **
5 Fine motor ASQ-3 area						0.445 **	0.224 **
6 Problem-solving ASQ-3 area							0.240 **
7 Personal–social ASQ-3 area							

* *p* < 0.05, ** *p* < 0.01.

**Table 3 children-10-00609-t003:** Model coefficient effect of maternal playfulness on each ASQ-3 area of children’s development.

	Mothers’ Intrusiveness			Communication			Gross Motor			Fine Motor			Problem-Solving			Personal–Social		
				Model 1			Model 2			Model 3			Model 4			Model 5		
	Coeff.	SE	*p*	Coeff.	SE	*p*	Coeff.	SE	*p*	Coeff.	SE	*p*	Coeff.	SE	*p*	Coeff.	SE	*p*
Maternal playfulness	−0.33	0.07	<0.001	0.14	0.08	0.08	0.11	0.08	0.19	0.07	0.08	0.37	0.07	0.08	0.36	0.24	0.08	0.004
Mothers’ intrusiveness	-	-	-	−0.21	0.11	0.07	−0.06	0.12	0.60	−0.22	0.12	0.06	−0.27	0.11	0.02	0.07	0.11	0.55
Intercept	3.39	0.22	<0.001	0.12	0.44	0.78	−0.16	0.47	0.73	0.34	0.45	0.46	0.45	0.45	0.31	−0.84	0.45	0.07
	R^2^ = 0.22			R^2^ = 0.14			R^2^ = 0.04			R^2^ = 0.09			R^2^ = 0.13			R^2^ = 0.11		
	*F* (1, 77) = 21.94, *p* < 0.001			*F* (2, 76) = 6.24, *p* = 0.003			*F* (2, 76) = 1.72, *p* = 0.185			*F* (2, 76) = 3.95, *p* = 0.023			*F* (2, 76) = 5.44, *p* = 0.006			*F* (2, 76) = 4.72, *p* = 0.011		

## Data Availability

Data are available upon request of the corresponding author.

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
