# Peer review of "Non-Intrusive Maternal Style as a Mediator between Playfulness and Children’s Development for Low-Income Chilean Adolescent Mothers"

_children, 2023, doi:10.3390/children10040609_

Round 1

Reviewer 1 Report

Thank you for the opportunity to read this very interesting study.  The work is an important addition to the literature and opens up further insights in cross cultural considerations. I am attaching a version ion adobe with comments on the paper. The main focus of my comments are around ensuring the methods are valid for the population and have been validated as such.  I hope my comments are useful.

Reviewer 2 Report

The paper addresses very important issues related to the association between parental characteristics and child's development. This study is particularly important since this association can significanly effect future developmental trends of a child.

Overall the paper is good, clearly presented, all the methods were described so that it would be absolutely clear how they were measured.

My only comment is related to conclusions and discussion. Your results showed that moderation role of mother's intrusiveness somewhere on the range from low to non-significant. But your describe it as 100%-prooved fact. Probably it would be better to use softer interpretations and add more limitations related to significance levels receaived in your study.

Reviewer 3 Report

Dear Authors,

Congratulations on your extensive work!

I suggest some minor revision:

Abstract:

 An ASQ-3 questionnaire was applied to measure the 16 children’s communication, gross and fine motor skills, problem-solving and personal-social development.

The authors should explain abbreviation ASQ-3.

M&M:

Those paragraphes should be divided into:

Study design, then inclusion and exclusion criteria,  then recrutiment to the study, then precedures,  then ethical issues, then study methods and finally  statistical analysis.

Lines 193-207: could be moved to Results section

Line 214: The eligibility criteria were as follows: mothers who 214 had become pregnant at 19 years old or younger, and children of typical development aged from 10 to 24 months. Could the authors explain how was the child’s development assessed during recrutiment to the study?

How about adding a participant flowchart?
